# LEVERAGING METAPATHS FOR LEARNING FROM KNOWLEDGE GRAPHS IN THE CONTEXT OF VISION-BASED CLASSIFICATION OF OBJECT STATES

## ABSTRACT

Zero-Shot Object State Classification (ZS-OSC) aims to recognize unseen object states without any visual training examples. Existing methods typically rely on Knowledge Graphs (KGs) to provide semantic information about states, but they often treat KGs as homogeneous, overlooking the rich relational knowledge encoded in their structure. We propose a novel approach to ZS-OSC[1] that leverages meta-paths to capture complex relationships between object states in a KG. Our method learns to project semantic information from the KG into the visual space via meta-path learning, generating discriminative visual embeddings for unseen state classes. To the best of our knowledge, this is the first work to utilize meta-paths for ZS-OSC. We conduct extensive experiments on four benchmark datasets, demonstrating the superior performance of our approach compared to SoTA zero-shot learning methods and a graph-based baseline. Our ablation study further provides insights into the impact of key design choices on the effectiveness of our method.

## 1 INTRODUCTION

Knowledge graphs (KGs) have become increasingly important in addressing various computer vision (CV) tasks, such as object classification (Marino et al., 2017), zero-shot recognition (Wang et al., 2018b) and Visual Question Answering (Krishna et al., 2017). This surge in KG utilization can be attributed to their ability to provide rich semantic information and contextual knowledge that can enhance the understanding of visual data. However, current approaches often under-utilize the full potential of KGs. Several factors contribute to this sub-optimal utilization. First, KGs used in Computer Vision (CV) are often large and contain irrelevant or erroneous information, which can introduce noise and hinder performance. Second, many methods fail to consider the diversity of edge types in KGs, treating all edges as homogeneous and overlooking valuable relational knowledge. This simplistic approach limits the ability to effectively exploit the rich semantic information embedded within KGs.

A number of approaches such as filtering mechanisms (Wang et al., 2014; Domingos & Richardson, 2007), ad-hoc KG construction (Dong et al., 2014; Gouidis et al., 2024) and random walking (Perozzi et al., 2014; Grover & Leskovec, 2016) were proposed as strategies against these shortcomings. An alternative approach focusing on the overcoming of these limitations concerned the concept of meta-paths (Sun et al., 2011; Dong et al., 2017). The utilization of meta-paths essentially involves the learning of the relative importance of the different paths between nodes within the KG via the assignment of weights to them. Meta-path-based approaches offer several advantages when applied to heterogeneous information networks (HINs). First, meta-paths enable the modeling of complex semantic relationships by explicitly defining sequences of node and edge types (Figure 1), making them particularly suitable for capturing the diverse interactions in multi-typed networks (Sun et al., 2011; Shi et al., 2014). This provides a significant advantage over traditional graph learning methods (Sun et al., 2011), which treat all nodes and edges as homogeneous, thus failing to leverage the rich information embedded in HINs. Furthermore, meta-paths allow for task-specific traversal and filtering, enabling more accurate representations for applications such as link prediction, recom-

---

[1]The code can be found at https://anonymous.4open.science/r/Metapaths-7811/

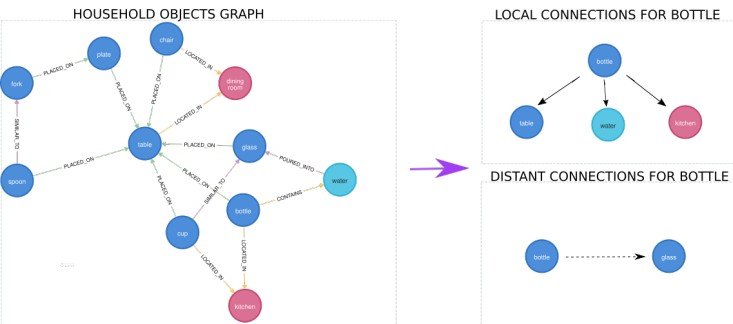

Figure 1: Unlike standard graph learning methods that prioritize local connections, meta-paths can capture distant relationships. In this toy example of a household objects graph, a meta-path can detect the stronger connection in specific contexts between "bottle" and "glass" (linked indirectly) than between "bottle" and "table", "bottle" and "kitchen" and "bottle" and "water" (directly linked).

mendation systems, and node classification. The ability to define domain-specific meta-paths also allows for more precise similarity measures, improving performance in tasks like clustering and search. Importantly, meta-paths enhance the interpretability of models by making the relationships between entities more transparent, providing domain experts with insights into how predictions are generated (Xiong et al., 2017).

Motivated by the great potential that the meta-paths learning seems to hold, this work attempts to explore the utilization of this approach in the context of the Object State Classification (OSC) task (Isola et al., 2015; Gouidis et al., 2022; Souček et al., 2022; Saini et al., 2023), which is a CV task attracting growing research attention over the last few years. OSC concerns the recognition of object states appearing in images and videos and is closely related to the more popular problems of Object Recognition and Action Recognition. OSC is an important problem whose solution is of significant impact. The recognition of object states and state changes is crucial for determining an object's condition and the interaction that was performed upon or could be performed in the future on it (Jamone et al., 2016). Moreover, the capacity for efficient OSC is of primary importance in AI systems that support tasks such as learning object affordances (Chuang et al., 2018), recognizing interactions (Wang et al., 2016b; Isola et al., 2015; Liu et al., 2017; Mancini et al., 2022), reasoning to achieve an object state change (Farhadi et al., 2009), recognizing the completion or failure of goals and recovery from possible mistakes during procedural activities (Schoonbeek et al., 2024) and many others. Meanwhile, large-scale video datasets (Grauman et al., 2022; Saini et al., 2023) concerning human-object interactions provide rich annotation data which are related to object state changes enabling the definition of new problems and the establishment of benchmarks and challenges related to object state detection and classification (Grauman et al., 2022).

This work addresses the challenging task of Zero-Shot Object State Classification (ZS-OSC), where the goal is to classify object states without any visual training examples. The key challenge lies in leveraging auxiliary non-visual information to enable the classification of these unseen states. Existing state-of-the-art methods typically utilize KGs as sources of structured semantic information (Gouidis et al., 2023), but they often treat KGs as homogeneous, potentially overlooking valuable relational information. We propose a novel approach that leverages meta-paths for more effective learning of zero-shot representations. Our method learns to project semantic information from a KG into the visual space via meta-path learning, generating visual embeddings for unseen state classes. To the best of our knowledge, this is the first method to utilize meta-paths in the context of Zero-Shot Visual Classification, and therefore ZS-OSC, through embedding generation.

Beyond their demonstrated utility in ZS-OSC, we posit that the generation of meta-path-based embeddings as a primary research objective holds significant promise. This direction offers several compelling advantages. First, meta-path-based embeddings can capture both local and global semantic relationships in heterogeneous networks, providing universal entity representations applicable across diverse tasks and domains without requiring task-specific fine-tuning. This fosters the development of multi-purpose, generalizable embeddings. Second, generating embeddings with a focus on representation quality can facilitate knowledge transfer across domains. Third, while cur-

rent embedding techniques, such as (Dong et al., 2017), are often task-specific, a framework that prioritizes the generation of meta-path embeddings as an independent objective would offer a generalized tool for analyzing heterogeneous information networks. This would mitigate the need for task-dependent tuning and enable researchers to investigate embeddings without being constrained by a particular task. Finally, since meta-paths effectively capture complex, higher-order relationships within networks, prioritizing the efficient generation of meta-path-based embeddings can lead to more compact and informative representations, optimizing storage and computational efficiency, particularly in large-scale heterogeneous networks.

This work makes the following key contributions:

- We introduce a novel method for generating embeddings by leveraging meta-paths within Graph Neural Networks (GNNs). This approach enables GNNs to effectively harness the rich information encoded in KGs. To the best of our knowledge, this is the first work to explore meta-path utilization for embedding generation in this context.
- We conduct a comprehensive ablation study to analyze the impact of various design choices and parameters on the performance of our proposed method. This analysis provides valuable insights into the interplay between meta-path learning and embedding generation.
- We perform an extensive experimental evaluation on four diverse datasets, comparing our method against established KG-based baselines and SoTA Large Pre-trained Models. The results demonstrate that our approach achieves superior performance by a significant margin.

## 2    RELATED WORK

**Meta-path Learning**: Meta-paths, a fundamental concept in heterogeneous information networks (HINs) (Shi et al., 2016), have been widely studied in applications such as similarity search, recommendation systems, and link prediction. Meta-paths were introduced to capture complex relationships in HINs, enabling the study of object proximities and connectivity patterns. The work by Sun et al. (2011) proposed PathSim that used meta-paths to measure similarity between objects based on shared relationships, with success in applications like similarity search and clustering. Extending this, Shi et al. (2014) proposed HeteSim, which computes relevance between objects via meta-paths, incorporating directionality and node types for enhanced flexibility. Meta-paths have also enhanced recommendation systems in HINs. Yu et al. (2013) developed a collaborative filtering algorithm incorporating meta-path-based similarities between users and items, improving recommendation accuracy. Similarly, Wang et al. (2016a) used meta-path-based features in matrix factorization for item recommendation, leveraging HIN structure to model user-item interactions more effectively.

Learning optimal meta-path weights for specific tasks has been a key research area. Dong et al. (2017) introduced MetaPath2Vec, which learns node embeddings through meta-path-based random walks, showing improved performance in classification and clustering tasks. Fu et al. (2020) extended this with a scalable meta-path-guided graph neural network, learning meta-path importance for tasks in large-scale HINs. In link prediction, meta-paths have been used to predict missing or future links in networks. Liu et al. (2018) proposed a meta-path-based link prediction method, capturing complex node interactions and outperforming traditional algorithms in networks with multiple types of nodes. Recent advancements in graph neural networks (GNNs) have further improved meta-path-based link prediction. Zhang et al. (2019) introduced a heterogeneous graph neural network (HGNN) that integrates meta-path-based features into node aggregation, achieving better link prediction.

Sun & Han (2013) introduced the concept of meta-paths for mining HINs, capturing semantic relationships across data types. Automatic discovery methods for meta-paths were later explored by Meng et al. (2015) to address challenges in manually retrieving meta-paths. Ferrini et al. (2024) proposed a novel approach to enhance GNN accuracy through effective meta-path identification, while Noori et al. (2023) explored meta-paths for flexible similarity search in biological knowledge graphs. Additionally, Yun et al. (2022) discussed learning new graph structures using meta-paths, demonstrating their role in enhancing GNN performance. Recent studies have also focused on challenges such as automatic meta-path discovery (Huang et al., 2020), integration with deep learning models (Wang et al., 2019), and the application of meta-paths in dynamic HINs (Trivedi et al., 2019).

The main novelty of our work distinguishing it from the aforementioned works concerns the utilization of meta-paths in the context of embeddings generation. Although many actual works focusing on meta-paths utilize embeddings, they serve as a means for another goal, i.e. downstream task, such as link prediction or entity alignment. In contrast, in our work, the generation of embeddings is a final objective.

**Object State and Attribute Classification**: Visual attributes are commonly defined as visual concepts that are both machine-detectable and human-understandable (Duan et al., 2012). The prevailing approach to learning attributes mirrors that of object classes, where a convolutional neural network is trained with discriminative classifiers using annotated image datasets (Singh & Lee, 2016). However, existing labeled attribute datasets suffer from limitations such as smaller scale compared to object datasets, a restricted number of generic attributes, and limited category coverage (Lampert et al., 2009; Isola et al., 2015; Patterson & Hays, 2016; Yu & Grauman, 2017; Mancini et al., 2022). Furthermore, research specifically focusing on state classification remains limited (Gouidis et al., 2022), with most existing approaches relying on similar assumptions and techniques as those employed for attribute classification. This highlights a need for dedicated methods tailored to the unique challenges of state recognition.

**Zero-shot Classification**: Zero-shot learning (ZSL) has attracted significant attention due to its ability to address the practical challenge of classifying objects without any training examples (Xian et al., 2018a). This is particularly crucial in real-world scenarios where obtaining labeled data for every possible class is often infeasible. Several approaches have been proposed for zero-shot object classification, including semantic embedding-based methods (Wang et al., 2018a; Xian et al., 2018b), attribute-based methods (Lampert et al., 2014), generative models (Xian et al., 2018b; Changpinyo et al., 2016), and learning compatibility functions between image and class embeddings (Akata et al., 2015). Semantic embedding methods utilize compact semantic spaces or attribute sets to bridge the gap between seen and unseen object classes. Attribute-based methods leverage descriptive attributes to infer the class of unseen objects. Generative models synthesize images of unseen classes based on similarities to seen classes. Additionally, recent work has explored the use of knowledge graphs to capture semantic relationships between objects and facilitate ZSL (Kampffmeyer et al., 2019; Nayak & Bach, 2022). While ZSL has been extensively studied for object recognition, its application to Zero State Classification (ZSC) remains relatively unexplored. With the exception of Gouidis et al. (2023), which focuses exclusively on state classification without relying on prior knowledge about object classes, existing ZSC methods primarily address the Compositional Zero-Shot Learning (CZSL) variant of the problem. CZSL aims to generalize to unseen combinations of object and state primitives by learning their compositionality from the training set (Misra et al., 2017; Nagarajan & Grauman, 2018; Yang et al., 2020).

## 3 METHODOLOGY

This work introduces a novel approach for generating embeddings in heterogeneous graphs by synergistically combining KGs structures with meta-path-based GNNs. These generated embeddings are then employed for the task of ZS-OSC. Our methodology harnesses the strengths of both KGs embeddings and the rich relational information encapsulated by meta-paths. The core idea is to leverage a designated set of KG nodes as guides for meta-path learning. Specifically, we utilize the visual embeddings of these nodes as ground truth and train a Graph Transformer Network (GTN) to assign weights to different KG edge types. This is achieved by generating embeddings for the guide nodes that align with their visual embeddings. The GTN architecture is inspired by Yun et al. (2022), while the training procedure draws inspiration from Kampffmeyer et al. (2019); Gouidis et al. (2023). Figure 2 provides a general overview of our method.

### 3.1 PRELIMINARIES

Before proceeding with describing our method, it is necessary to present related backgrounds related to our work. Additionally, Table S1 presented in the supplementary section lists commonly used notations in this paper for quick reference.

**Heterogeneous Graph**: A heterogeneous graph is denoted as $G = (V, E, \phi, \psi)$ and is associated with a node type mapping function $\phi : V \to T_v$ and an edge type mapping function $\psi : E \to T_e$,

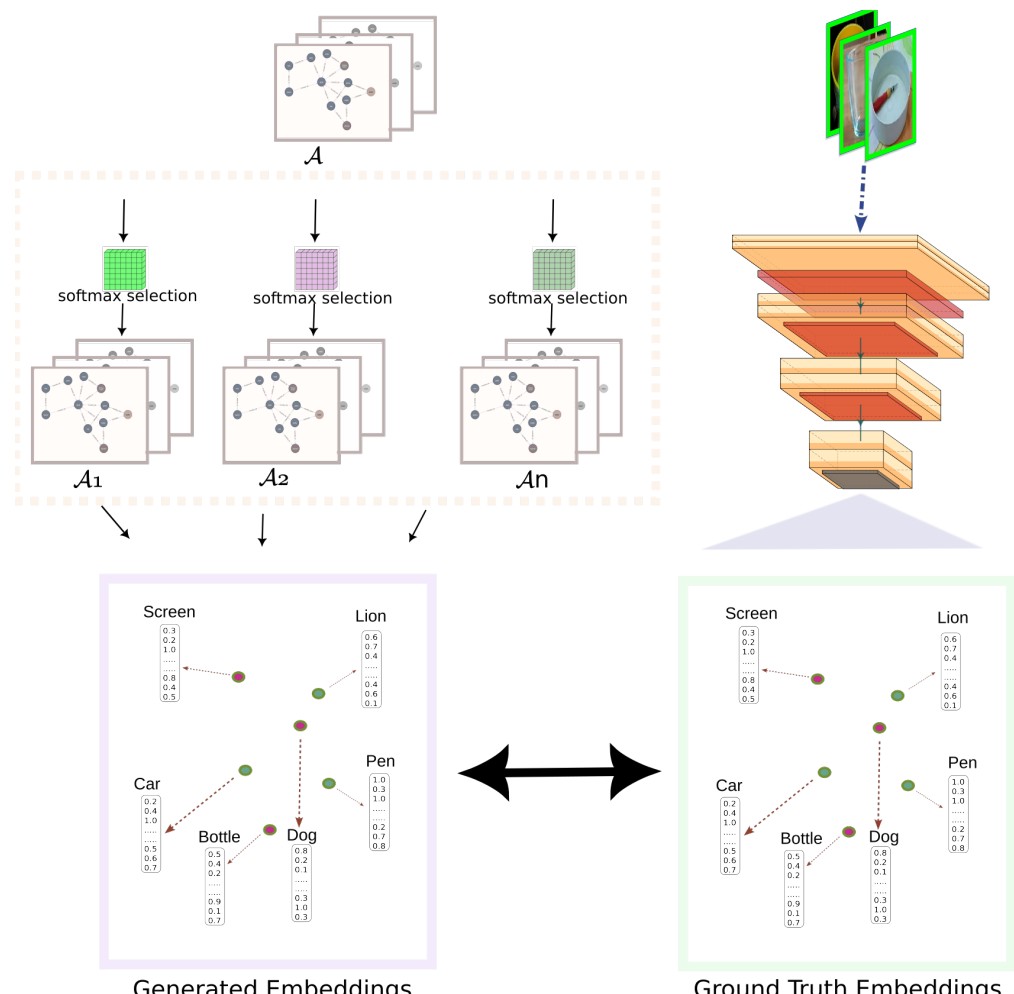

Figure 2: Overview of the proposed approach. A Graph Transformer Network (GTN) with a softmax selection mechanism assigns weights to different meta-paths. Meta-paths are aggregated and combined with node features to generate node embeddings. Visual ground truth embeddings for a set of guide nodes are produced using a pre-trained convolutional neural network. These embeddings guide the training of the GTN.

where $T_v$ and $T_e$ denote the set of nodes types and edges types respectively. A necessary requirement for a graph to be considered heterogeneous is to contained more than one type of nodes or more than one type of edges, i.e., $|T_v| + |T_e| > 2$.

**Metapath**: Within heterogeneous graphs, a meta-path represents a sequence of connected nodes traversed through diverse edge types. Formally, it can be defined as a path:

$$v_1 \xrightarrow{\psi(e_1)} v_2 \xrightarrow{\psi(e_2)} \dots \xrightarrow{\psi(e_\ell)} v_{\ell+1},$$

where each edge $e_\ell$ in the path has a corresponding edge type $\tau_e(e_\ell)$, belonging to the set of all possible edge types $T_e$.

**Metapath Instance**: A metapath instance $p$ of a metapath $P$ in a heterogeneous graph is defined as a sequence of nodes that follows the schema defined by $P$.

**Metapath-based Neighbor**: For a node $v$ and a metapath $P$ in a heterogeneous graph, the set of metapath-based neighbors $N_P(v)$ is defined as the set of nodes connected to $v$ via metapath instances

of $P$. A neighbor that is connected through two different metapath instances is treated as two distinct nodes in $N_P(v)$. If $P$ is symmetric, $N_P(v)$ includes the node $v$ itself.

**Metapath-based Graph**: For a given metapath $P$, the metapath-based graph $G_P$ is the graph constructed using all metapath-based neighbor pairs from the original graph $G$. If $P$ is symmetric, $G_P$ is homogeneous.

**Heterogeneous Graph Embedding**: Given a heterogeneous graph $G = (V, E)$, with node attribute matrices $X_{A_i} \in \mathbb{R}^{|V_{A_i}| \times d_{A_i}}$ for each node type $A_i \in \mathcal{A}$, the goal of heterogeneous graph embedding is to learn $d$-dimensional node representations $h_v \in \mathbb{R}^d$ for each node $v \in V$, where $d \ll |V|$, capturing rich structural and semantic information from $G$.

### 3.2 META-PATH-BASED EMBEDDING GENERATION

Meta-paths represent semantic connections between different node types in a heterogeneous graph. GTNs automatically learn useful meta-paths by selecting and combining adjacency matrices of different edge types. This process allows the model to generate new graph structures that are useful and is used typically for downstream tasks such as node classification. In our case, the meta-paths learning is not mediated by a downstream task. Instead, meta-paths are learned via the generation of embeddings for the graph's nodes.

Following the notation introduced previously, we can use adjacency matrices, a different one for each edge type, to compute the different meta-paths. Specifically, the meta-path's adjacency matrix $A_P \in \{0, 1\}^{\mathbb{V} \times \mathbb{V}}$ of the graph is computed as:

$$A_P = \prod_{i=1}^{\ell} A_{t_i} \implies A_P = A_{t_1} A_{t_2} \ldots A_{t_\ell},$$

where $A_{t_i}$ is the adjacency matrix corresponding to edge type $t_i$ and $l$ corresponds to the length of the meta-paths, with $a_{ij} = 1$ denoting a meta-path with length equal than $l$ between nodes $i$ and $j$ and $a_{ij} = 0$ an absence of meta-path, respectively. In order to have also meta-paths with lengths less than $l$ we add the identity matrix $I \in \mathbb{R}^{\mathcal{V} \times \mathcal{V}}$ to the adjacency matrices.

With the utilization of a soft selection mechanism, weights are assigned to the different types of edges. This technique enables the learning of the optimal combination of edge types for each meta-path. Specifically, this is achieved with a $1 \times 1$ convolution with softmax activation over the adjacency matrices of different edge types:

$$WA^{(k)} = \sum_{t=1}^{|T_e|} \alpha_t^{(k)} A_t,$$

where $\alpha_t^{(k)}$ is the learnt weight for edge type $t$ at the $k$-th transformer layer. The resulting meta-path adjacency matrices are normalized using degree matrices $D$ and applied to perform multi-hop convolutions. More than one softmax convolutions could be also applied, in which case the objective is that each channel learns a different representation and the overall combination of the different representations results in more a robust outcome.

With the learnt softmax weights, the meta-path's adjacency matrix $WA_P \in \mathbb{R}^{\mathbb{V} \times \mathbb{V}}$ of the graph is computed as:

$$WA_P = \prod_{i=1}^{\ell} A_{t_i} \implies \sum_{t=0}^{|T_e|} \alpha_t^{(k)} A_t = \sum_{t=1}^{|T_e|} \alpha^{(0)} A_{t_0} \sum_{t=1}^{|T_e|} \alpha^{(1)} A_{t_1} \sum_{t=1}^{|T_e|} \alpha^{(2)} A_{t_2} \cdots \sum_{t=1}^{|T_e|} \alpha^{(l)} A_{t_l}.$$

After having computed the weighted adjacency matrix $WA_P$ we can compute the aggregation meta-paths-based features matrix $\mathbb{X}_P \in \mathbb{R}^{|V_{A_i}| \times d_{A_i}}$ for the graphs nodes by multiplying the nodes features matrix matrices $\mathbb{X}_{A_i} \in \mathbb{R}^{|V_{A_i}| \times d_{A_i}}$ with $WA_P$:

$$\mathbb{X}_P = \mathbb{X}_A \times WA_P.$$

In order to generate embeddings for the graphs nodes we use a multi-layer Graph Neural Network which learns to project the nodes features into the target space:

$$\mathbb{E}_P = f_\theta(\mathbb{X}_P).$$

The weights corresponding to the softmax layers and the stacked layers of the GNN are updated based on the minimization of a L2 distance function $\mathcal{L}_{\mathcal{E}}$ between the generated nodes embeddings and the ground truth nodes embeddings for the set of nodes which serve as ground truths:

$$\mathcal{L}_{\mathcal{E}} = \frac{1}{2N} \sum_{n \in \mathcal{N}} \sum_{d \in \mathcal{D}} (\mathbb{E}_P - \tilde{\mathbb{E}}_P)^2,$$

where $\mathcal{D}$ is the dimension of the embeddings and $\mathcal{N}$ the number of the ground truth concepts, respectively.

### 3.3 Zero-Shot Object State Classification

To achieve zero-shot object-state classification, it is crucial to learn representations for the unseen target state classes. To this end, we construct a KG with state classes as nodes and utilize the GTN trained in the previous step for meta-path learning to generate embeddings for these target classes. These embeddings are then integrated into a visual classifier, specifically a pre-trained CNN, to serve as visual representations of the visually known state classes. Following established practices in transfer learning and zero-shot learning, we replace the final layer of the pre-trained CNN with these generated embeddings. To ensure compatibility, the dimensionality ($\mathcal{D}$) of the generated embeddings is set to match the dimension of the CN's last layer. This adaptation enables the CNN to effectively classify the target state classes in a zero-shot manner.

## 4 Experimental Evaluation

### 4.1 Implementation and evaluation issues

**Implementation Details:** The KG provided as input to the Graph Transformer Network (GTN) for meta-path learning was the WordNet hierarchy of the 1000 classes from the ImageNet1000 dataset (Russakovsky et al., 2015). These 1000 classes served as the ground truth set for training the GTN, which was trained for 200 epochs with five different learning rates (see Section 4.2 for details). The same KG was also used to generate the target object state classes. In experiments with multiple softmax channels, the generated embeddings were averaged. The convolutional neural network (CNN) used for zero-shot object-state classification (ZS-OSC) was ResNet-101 (He et al., 2016) pre-trained on the ImageNet1000 dataset.

**Datasets:** We utilized four publicly available datasets containing object state annotations: OSDD Gouidis et al. (2022), CGQA Mancini et al. (2022), MIT Isola et al. (2015), and VAW Pham et al. (2021). While OSDD is specifically designed for state detection, the other three are attribute datasets that include object states among their classes. We extracted subsets concerning exclusively object state classes. For OSDD and VAW, bounding boxes from the original images were extracted to create images suitable for the OSC task. Table S2 in the supplementary section provides details on the four datasets.

**Metrics**: Our evaluation follows the zero-shot method from (Purushwalkam et al., 2019). Following those guidelines we calculate accuracy per class and then average these instead of reporting overall accuracy. This ensures each class is equally important, regardless of its sample size.

### 4.2 Ablation Study

This ablation study investigates the optimal configuration of key parameters related to meta-path-based embedding generation, including: (a) the maximum length of meta-paths, (b) the number of softmax channels used for meta-path selection, and (c) the training learning rate.

| Meta-paths Max Length | OSDD | CGQA-States | MIT-States | VAW-States |
|:---:|:---:|:---:|:---:|:---:|
| 1 | 27.1 | 44.9 | 47.1 | 29.3 |
| 2 | 29.5 | 46.7 | 47.3 | 29.8 |
| 3 | **31.3** | **47.6** | **48.7** | **32.1** |
| 4 | 30.2 | 46.5 | 47.9 | 31.0 |

Table 1: Ablation results for maximum length of meta-paths. The number of softmax channels in the GTN was 3. The networks were trained with a learning rate equal to $1e-2$.

| Number of Channels | OSDD | CGQA-States | MIT-States | VAW-States |
|:---:|:---:|:---:|:---:|:---:|
| 1 | 27.2 | 44.3 | 46.2 | 28.5 |
| 2 | 28.9 | 46.1 | 46.4 | 29.7 |
| 3 | **31.3** | **47.6** | **48.7** | **32.1** |
| 4 | 30.5 | 46.9 | 47.0 | 30.7 |

Table 2: Ablation results for number of softmax channels. The maximum length of the meta-paths was 3. The networks were trained with a learning rate equal to $1e-2$.

**Length of meta-path:** Table 1 presents the results of varying the maximum length of meta-paths (1, 2, 3, and 4 hops). The best performance across all datasets was achieved with a maximum length of 3 hops. Performance generally improved with increasing meta-path length, likely because longer paths capture more global graph information. However, performance slightly decreased with a length of 4 hops, suggesting that excessively long paths might introduce noise, potentially by incorporating less relevant or spurious relationships. This finding highlights the importance of carefully selecting the appropriate meta-path length to balance information gain and noise reduction.

**Number of Softmax Channels**: Table 2 shows the impact of varying the number of softmax channels used for meta-path selection. The best performance was achieved with 3 channels, suggesting that multiple channels allow the GTN to learn diverse meta-path representations, thereby improving embedding quality. Increasing the number of channels allows the model to capture different aspects of the relationships encoded in the meta-paths, leading to richer and more informative embeddings. However, using too many channels, e.g., 4 in this case, might not provide further benefits and could potentially increase model complexity without a corresponding improvement in performance.

**Training Learning Rate:** Table 3 presents the results of using different learning rates (LR) during GTN training. The best performance was obtained with a learning rate of 1e-2. Smaller learning rates led to a significant performance drop, indicating the importance of this parameter for effective model training. This suggests that a learning rate of 1e-2 strikes a good balance between convergence speed and stability, allowing the model to effectively learn from the data without overshooting or getting stuck in local optima. The observed performance drop with smaller learning rates could be attributed to slower convergence and potential difficulties in escaping local optima.

### 4.3 COMPARISON TO COMPETING METHODS

This experiment had a two-fold objective. First, we compared our method against SoTA Large Pre-trained Models (LPMs) capable of ZS-OSC. We used six different prompts[2] related to the target states and report the mean average performance across all prompts, following standard convention. Second, we compared our approach to a graph-based method (Gouidis et al., 2023) specifically designed for ZS-OSC, which allows us to assess the impact of meta-paths on performance. For comparison with the baseline method (Gouidis et al., 2023), which relies on random walks, we used five different random seeds for initialization and report the mean performance over all seeds. The same KG was used as input to both the GCNs of the baseline method and our method. It is important to note that other ZSL methods, such as CZSL, are not applicable in this context, because they require information about object classes, which is not available in this zero-shot setting.

---

[2]The prompts are presented in the supplementary material.

| LR | OSDD | CGQA-States | MIT-States | VAW-States |
|---|---|---|---|---|
| $5e-2$ | 29.2 | 46.2 | 47.5 | 30.1 |
| $1e-2$ | **31.3** | **47.6** | **48.7** | **32.1** |
| $5e-3$ | 28.9 | 45.1 | 47.0 | 28.5 |
| $1e-3$ | 27.2 | 44.5 | 46.1 | 26.2 |
| $5e-4$ | 26.1 | 43.8 | 44.5 | 24.6 |

Table 3: Ablation results for learning rate. The maximum length of the meta-paths was 3, and the number of softmax channels was 3.

| Method | OSDD | CGQA-States | MIT-States | VAW-States |
|---|---|---|---|---|
| Baseline (Gouidis et al., 2023) | 27.3 | 45.1 | 43.3 | 25.6 |
| CLIP-RN101 (Radford et al., 2021) | 22.5 | 46.9 | 39.3 | 28.0 |
| CLIP-VITBP16 (Radford et al., 2021) | 28.8 | 44.9 | 46.4 | 30.1 |
| CLIP-VITLP14 (Radford et al., 2021) | 28.4 | 43.4 | 48.6 | 27.9 |
| ALIGN (Jia et al., 2021) | 29.5 | 40.0 | 44.2 | 28.4 |
| BLIP (Li et al., 2022) | 13.3 | 26.0 | 27.2 | 16.1 |
| **Ours** | **31.3** | **47.6** | **48.7** | **32.1** |
| Our improvement over the SoTA | 1.8 | 0.7 | 0.1 | 2.0 |
| Our improvement over the Baseline | 4.0 | 2.4 | 5.4 | 6.5 |

Table 4: Results against competing methods. Bold/Blue text indicates best/2nd best performance.

Table 4 presents the results. Our method achieved the best performance across all datasets, outperforming both the baseline method and all LPMs. Notably, our method surpassed the baseline by a large margin (4.0%, 2.5%, 5.4%, and 6.5% for OSDD, CGQA, MIT, and VAW, respectively).This outcome highlights the effectiveness of meta-path learning compared to random walks for representation learning in graph structures. The superior performance of our method can be attributed to the ability of meta-paths to capture and exploit specific semantic relationships within the KG, leading to more informative and discriminative embeddings for ZS-OSC. In contrast, random walk techniques might not effectively capture these relationships, resulting in less effective representations.

Furthermore, the fact that our method outperforms SoTA LPMs (1.8%, 0.7%, 0.1%, and 2.0% for OSDD, CGQA, MIT, and VAW, respectively) demonstrates the potential of incorporating KG information and meta-path learning into ZS-OSC. LPMs, while powerful, might not fully capture the nuanced semantic relationships between objects and their states that are encoded in KGs. By leveraging meta-paths to extract and utilize this information, our method achieves a significant performance improvement.

## 5 CONCLUSION

This paper introduced a novel method for generating embeddings in heterogeneous graphs by leveraging meta-paths within a graph neural network framework. Our approach utilizes KGs structures and visual embeddings to guide the learning of meta-paths, enabling the generation of informative and discriminative embeddings. We demonstrated the effectiveness of our method in the context of ZS-OSC, achieving superior performance compared to state-of-the-art LPMs and a graph-based baseline. Our ablation study provided insights into the impact of key parameters on the performance of our method.

Future work will focus on several promising directions. First, we aim to explore the applicability of our method to other zero shot CV tasks such as semantic segmentation, image captioning and visual question answering to further evaluate its generalizability. Second, we plan to investigate the integration of different types of KGs such as multimodal KGs and KGs containing different type of nodes and explore the impact of KG quality on embedding generation. Finally, we intend to extend our approach to incorporate more complex meta-path structures and explore alternative graph neural network architectures for enhanced performance.

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

# 6 SUPPLEMENTARY MATERIAL

This section presents tables that due to space limitation were omitted from the main body of the paper. Specifically, Table S1 shows the notation symbols that are used, Table S2 presents the details about the datasets used in the experimental evaluation and Table S3 shows the prompts that were using for the LPMs respectively.

| Symbol | Description |
|---|---|
| $G = (V, E)$ | Heterogeneous graph |
| $V$ | Set of nodes in $G$ |
| $E$ | Set of edges in $G$ |
| $\phi : V \to T_v$ | Node type mapping function |
| $\psi : E \to T_e$ | Edge type mapping function |
| $T_v$ | Set of node types |
| $T_e$ | Set of edge types |
| $|T_v|$ | Number of node types |
| $|T_e|$ | Number of edge types |
| P | Meta-path |
| $p$ | Meta-path instance |
| $v_i$ | Node in $G$ |
| $e_l$ | Edge in $G$ |
| $\tau_e(e_l)$ | Edge type of edge $e_l$ |
| $N_P(v)$ | Set of metapath-based neighbors of node $v$ under meta-path $P$ |
| $G_P$ | Metapath-based graph derived from meta-path $P$ |
| $X_A$ | Node feature matrix |
| $\mathcal{A}$ | Adjacency matrix |
| $\mathcal{A}_\mathcal{P}$ | Meta-path adjacency matrix |
| $\mathbb{E}_P$ | Meta-path based generated embedding of node $v$ |
| $\tilde{\mathbb{E}}_P$ | Ground truth embedding of node $v$ |
| $\mathcal{D}$ | Dimensionality of node embeddings |
| $\mathcal{N}$ | Number of ground truth concepts |
| $\mathcal{L}_\mathcal{E}$ | L2 distance loss function |

Table S1: Notation Table

| Dataset | Train | Val | Test | States | Objects | VOSC | TOSC | S\O |
|---|---|---|---|---|---|---|---|---|
| OSDD (Gouidis et al., 2022) | 6,977 | 1,124 | 5,275 | 9 | 14 | 35 | 126 | 2.36 |
| CGQA-states (Mancini et al., 2022) | 244 | 46 | 806 | 5 | 17 | 41 | 75 | 1.71 |
| MIT-states (Isola et al., 2015) | 170 | 34 | 274 | 5 | 14 | 20 | 70 | 1.57 |
| VAW (Pham et al., 2021) | 2,752 | 516 | 1,584 | 9 | 23 | 51 | 207 | 2.61 |

Table S2: Details about the four image datasets utilized in this work. Train/Val/Test: Number of Training/Validation/Testing Images. States: Number of State classes, Objects: Number of Object classes. VOSC/TOSC: Valid/Total Object-State combinations. S\O: Average number of states than an Object can be situated in.

| | Prompt |
|---|---|
| 1 | An image of a {} object |
| 2 | The object in the image is {} |
| 3 | The state of the object in the image is {} |
| 4 | The object in the image is currently {} |
| 5 | An image of a object in a state of {} |
| 6 | The scene depicts a object that appears to be {} |

Table S3: Prompts used for the Large Pretained Models.

