# OpenReview forum: "Leveraging metapaths for learning from knowledge graphs in the context of vision-based classification of object states"
_ICLR.cc/2025/Conference — ICLR 2025 Conference Withdrawn Submission_

### Official Review · Reviewer_EVC4 · 2024-10-30

**Soundness:** 2
**Presentation:** 2
**Contribution:** 2
**Rating:** 5
**Confidence:** 4

**Summary:**

This paper proposes a  Zero-Shot Object State Classification (ZS-OSC) method by leveraging metapaths in knowledge graphs (KGs) to capture and utilize the rich relational information encoded within their structure. The proposed method aims to recognize unseen object states without any visual training examples, addressing a key challenge in zero-shot learning. Extensive experiments conducted on four benchmark datasets demonstrate that the proposed approach outperforms a baseline method.

**Strengths:**

+ Leveraging meta-paths to capture complex relationships to learn a zero-shot representation seems reasonable.
+ The experimental results show the effectiveness of the proposed approach on conventional ZS-OSC.

**Weaknesses:**

- The related work section has a high degree of overlap with the baseline method [r1], especially in the "Zero-shot Classification" part. Additionally, there is a lack of in-depth analysis regarding ZSL, and the references are not up-to-date.
- The paper lacks experiments in a generalized ZS-OSC setting, which is more challenging and more reflective of real-world scenarios. [r1] performed the experiments.


[r1] Leveraging knowledge graphs for zero-shot object-agnostic state classification.

**Questions:**

- How were the experimental results for the other methods in Table 4 generated? Why do the vision-language model-based methods perform worse than the CNN-based method proposed in this paper?
- What are the practical applications of ZS-OSC? How does it differ from classical zero-shot learning or compositional zero-shot learning?
- Line 349, 'CN‘s' is so bad.
- Please follow the ICLR  standards for mathematical notations, such as Line 325.

---

> ### Author Response · Authors · 2024-12-03
> **Response to Reviewer EVC4**
>
> REVIEWER 5
>
> We thank the reviewer for their valuable feedback and address the raised concerns as follows.
>
> **Issues with related work:** We plan to update the related work section including some more recent works in case of acceptance. However, it is important to note that none of this work is essential for the problem of zero-shot OSC.
>
> **Experimentation in the Generalized setting:** The inclusion of the generalized ZS-OSC scenario is something we aim to pursue in future work. However, we deliberately chose not to include this type of experimentation in the current study for several reasons. First, a critical aspect of the generalized variation is the refinement of methods using training samples, which introduces additional complexity compared to pure zero-shot approaches. This makes generalized zero-shot learning (GSZ) inherently more difficult to interpret, especially when isolating the specific contribution of individual components. Second, testing our method in the GSZ-OSC scenario would introduce confounding factors, as the refinement process significantly influences final performance.
>
> **Reported results in the experimental evaluation section:**
>
> **1. How Were the Experimental Results for the Other Methods Generated?**
>
> The results for the five vision-language models listed in Table 4 were generated using their publicly available implementations on **Hugging Face**. For consistency, we evaluated these models on the same test dataset splits used in our experiments. Regarding the reported performance, we employed 6 different prompts for each method and reported the average accuracy.
>
> **2. Why Do Vision-Language Models Perform Worse Than the Proposed CNN-Based Method?**
> While vision-language models  are powerful and have demonstrated success in many zero-shot tasks, their underperformance in ZS-OSC is due to the specific nature of the problem:
> - Lack of State-Specific Training: 	Vision-language models are pre-trained on large-scale datasets with general-purpose visual and textual descriptions. However, these datasets often lack fine-grained state-specific annotations.
>
> - Inability to Leverage Structured Knowledge Graphs: Vision-language models do not inherently utilize structured relational information, such as that provided by KGs. Our method’s reliance on KGs and meta-paths allows it to incorporate fine-grained semantic relationships that vision-language models cannot access or represent.
> - State Generalization Limitations: Vision-language models often conflate object attributes with states, which can reduce their effectiveness in tasks that require distinguishing between specific states across diverse objects.
> - Advantages 	of Task-Specific Design: Our CNN-based method is specifically designed for ZS-OSC, leveraging:
>    - A GNN trained on KGs with meta-path learning to generate state-specific embeddings.
>     -  Visual embeddings fine-tuned for the task, which are aligned with the semantic embeddings to ensure compatibility.
> These factors enable our approach to outperform general-purpose vision-language models in this specialized task.
>
> **Practical applications of ZS-OSC and differences with classical ZSL and CZSL:**
>
> **1. Practical Applications of Obj-agn ZS-OSC**
> - Robotics and Automation: In robotics, the ability to recognize unseen object states  is 	essential for tasks such as object manipulation, assembly, and quality control.
> - Assistive Technologies: Devices designed for visually impaired individuals can benefit from Obj-agn ZS-OSC by providing contextual information about the environment.
> - Human-Object Interaction  Understanding: In applications such as surveillance or augmented reality, recognizing object states  can enhance scene understanding and facilitate human-object interaction analysis.
> - Scene 	Understanding in Videos: Obj-agn ZS-OSC plays a crucial role in video understanding by detecting 	object state changes over time, which is important in applications 	like procedural task analysis, monitoring systems, or action recognition.
>
> **2. How ZS-OSC Differs from Classical ZSL and CZSL**
>
> Obj-agn ZS-OSC is a specific and challenging variation of zero-shot learning, distinct from both classical ZSL and CZSL in the following ways:
> - Focus on Object States, Not Categories:
>    - ZSL: Focuses on recognizing unseen object categories by leveraging auxiliary knowledge, such as attributes or semantic  embeddings.
>     - Obj-agn ZS-OSC: Focuses solely on the states of objects, such as "open or " "empty," regardless of the object’s category.
>
> - Decoupling Object and State Classification:
>     - CZSL: Relies on the compositionality of objects and states  and assumes that the relationships  between objects and states seen during training can generalize to new combinations.
>     -  Obj-agn ZS-OSC: Decouples the classification of states from the classification of objects. This allows for recognizing states even when the object is unknown.

---

### Official Review · Reviewer_B9Bp · 2024-11-01

**Soundness:** 2
**Presentation:** 2
**Contribution:** 2
**Rating:** 5
**Confidence:** 4

**Summary:**

In this work, the authors propose a new method for ZS-OSC that utilizes meta-paths to capture complex relationships between object states in KGs, and then projects the semantic information in KGs into the visual space to generate visual embeddings for unseen state categories for zero-shot object state classification. The method in this paper shows competitive results on four datasets.

**Strengths:**

+ The structure of the article is clear and logical. Especially in the experimental part, the authors clearly show the results of the ablation experiments and the corresponding analysis.

+ The proposed method shows excellent performance in several datasets. For example, it outperforms the baseline method by 5.4% and 6.5% on the MIT-States and VAW-States datasets, respectively.

**Weaknesses:**

- In the related work, the authors focus on the current state of research and challenges in attribute classification, while the research on state classification is skipped. We are curious about the detailed analysis of the current state of research and key challenges in state classification, otherwise, this would not justify the author's statement “This highlights a need for dedicated methods tailored to the unique challenges of state recognition”.

- As the authors note, existing ZSC methods primarily address a CZSL variant of the problem, which generalizes to unseen combinations of object and state primitives by learning the compositionality of object and state primitives from the training set. This approach is clearly more reasonable than focusing only on state classification, since isn't state classification ultimately in the service of object classification? Could the authors please clarify what is the point of focusing only on state categorization? Alternatively, are the authors trying to show that state categorization can be achieved without relying on prior knowledge of object categories? Could the authors please focus on the uniqueness of this study that distinguishes it from existing studies on ZSC, other than the introduction of meta-path learning?

- The authors are advised to elaborate on the meta-path selection process, including how meta-paths are identified based on the structure of the knowledge graph and task requirements, and how the effectiveness of different meta-paths is evaluated. The authors mention in the conclusion that the approach in this paper utilizes the structure of KGs and visual embedding to guide the learning of meta-paths, but this is not clearly reflected in the methods section.

- The authors' key rationale for introducing meta-path learning into zero-shot object state classification is that meta-path learning can effectively capture and utilize specific semantic relationships in KGs, but there is no analysis of the rationale and a lack of corresponding visual representation. In addition, I can't see the changes in the features of the GTN trained based on meta-path learning when it is used in object state classification. Case studies demonstrating how meta-path learning helps in classifying challenging examples.

- Figures are less readable. For example, the font in Figure 1 is too small to read.

**Questions:**

All the doubts are mentioned in the weaknesses. It should be emphasized that the authors should focus on explaining the rationale of metapaths for capturing complex semantic information. In addition, authors should prove in more forms that the introduction of meta-path learning indeed produces “informative and discriminative visual embeddings”, as this is the core point of the paper.

---

> ### Author Response · Authors · 2024-12-03
> **Response to Reviewer B9Bp**
>
> We thank the reviewer for their valuable feedback and address the raised concerns as follows.
>
> **Issues with the related work section:** Thank you for pointing out the need to provide a more detailed analysis of the current state of research and key challenges in state classification to support our statement about the unique challenges of state recognition. We plan to include a dedicated subsection in the revised version of the paper in case of acceptance.
>
> **The Object-agnostic variation of the problem w.r.t. ZSC and CSZL:**  We would like to clarify that our work does not disregard the utility of object class knowledge for state classification. Instead, we argue that focusing solely on state classification, without relying on the accurate identification of object classes, offers several distinct advantages, particularly in real-world and zero-shot scenarios:
> - Robustness 	in Real-World Scenarios:
>  Object-agnostic state classifiers are inherently more robust in real-world settings where objects may be unknown, novel, or challenging to classify  and/or where object classifiers may fail or produce incorrect labels.
> - Recognition 	Across Diverse Objects:
>  By not relying on object-specific features, our method can generalize states across visually dissimilar objects.  An object-agnostic approach ensures that the classifier focuses solely on state-relevant cues rather than object-specific visual features, which improves its ability to generalize.
>
> **Practical Applications of Object-Agnostic State Classification:**
> - Human-Object Interaction: Many applications, such as robotics and assistive technologies, prioritize understanding the state of an object (e.g., "closed," "filled") over its precise identity, as the object's state often dictates the required action (e.g., grasping, pouring).
> - Dynamic Environments: In scenarios where object categories are dynamic or undefined (e.g., identifying the state of novel tools in a factory), state classification independent of object knowledge allows the system to adapt to new tasks or objects.
> - Advancing the Zero-Shot Paradigm:
>  Most existing zero-shot learning methods in the compositional ZSL (CZSL) domain rely on learning the compositionality of object and 	state primitives (e.g., "red car" or "wooden chair") from seen combinations. While effective, these methods inherently require prior knowledge of object categories. By focusing purely on state classification, our work takes a step toward object-agnostic 	zero-shot learning, 	which is a more general and challenging problem.
> - Uniqueness of Our Study Beyond Meta-Path Learning:
> Our study highlights the feasibility and benefits of object-agnostic state classification. This differentiates our work from existing studies, which often approach state recognition as a secondary task within object-centric 	frameworks.
>
>
>  **Elaboration on the meta-path selection process:** The objective of our method is to project semantic representations into the visual space to enable zero-shot classification effectively. In our framework, we use the visual embeddings of a set of known classes as ground truth representations in the visual space. Each of these classes corresponds to a node in the KG, which is enriched with semantic features. Using this setup, we train a GNN to generate embeddings for all KG nodes. The training loss is computed by comparing the generated embeddings to the visual ground truths.
>
> The structure of the KG plays a critical role in determining the quality of the generated embeddings. The GNN aggregates information from each node and its neighbors, making it sensitive to the graph's topology and the relationships encoded within it. However, standard aggregation treat all relationships equally. To address this, we leverage meta-paths to enhance the embedding generation process. Meta-paths define specific sequences of node and edge types within the KG, allowing us to explicitly model the semantic relationships most relevant to our task. By assigning different weights to the relations along these meta-paths, the model prioritizes task-relevant connections, refining the aggregation process and enabling the generation of higher-quality embeddings.
>
> The meta-path selection process is guided by both the structure of the KG and the requirements of the ZS-OSC task. Specifically:
> - Task Relevance: Meta-paths are designed to capture the semantic pathways that align most closely with the classification objective.
> - Automatic Learning of Weights: Through our softmax-based selection mechanism within the GNN, the model learns to assign optimal weights to different edge types during training. This enables the automatic identification of the most informative meta-paths without requiring manual tuning.
> - Evaluation of Effectiveness: The effectiveness of different meta-paths is inherently evaluated during training, as their contribution is reflected in the model's ability to generate embeddings that align with the visual ground truths.

---

### Official Review · Reviewer_3FaH · 2024-11-01

**Soundness:** 2
**Presentation:** 1
**Contribution:** 2
**Rating:** 3
**Confidence:** 3

**Summary:**

This paper delves into Zero-Shot Object State Classification, introducing meta-path learning to enable heterogeneous modeling within a knowledge graph. Unlike existing approaches that focus solely on homogeneous modeling, this work showcases the effectiveness of the proposed methodology through experimental validation.

**Strengths:**

The incorporation of heterogeneous modeling is a pivotal step forward in object state classification and zero-shot object state classification. This innovative approach represents a promising direction for advancing the field.

**Weaknesses:**

It appears that the clarity of the paper is a critical issue, hindering comprehension even for experienced researchers in the field of computer vision. Improving the accessibility and clarity of the manuscript is crucial before considering it for publication. Here are some key areas for improvement:
1. The authors should dedicate space to formally define the zero-shot Object State Classification problem within the paper. Given its relevance in the current landscape, a comprehensive definition in the methodology section is essential. Additionally, providing detailed explanations about Object State Classification, including its inputs and outputs, would enhance understanding.
2. Although the paper elaborates on learning a meta-path-based graph network, it falls short in clearly articulating how this network is applied in object state classification and zero-shot object state classification.
3. While heterogeneous modeling is deemed critical for object state classification, the paper lacks substantial evidence to support the assertion that meta-path learning effectively introduces heterogeneous information. The motivation behind choosing meta-path learning over other techniques remains insufficiently justified. Including experimental comparisons with alternative methods could strengthen the argument.
4. The authors should address why meta-path learning specifically benefits zero-shot Object State Classification. While heterogeneous modeling is crucial for object state classification, the connection between meta-path learning and the zero-shot context needs clarification.

**Questions:**

In addition to addressing the above points, it is advisable for the authors to conduct a thorough limitation analysis and present qualitative results to enhance the comprehensiveness of their work.

---

> ### Author Response · Authors · 2024-12-03
> **Response to Reviewer 3FaH**
>
> We thank the reviewer for their valuable feedback and address the raised concerns as follows.
>
> **Formulation of the zero-shot Object State Classification task:** We agree that providing a formal definition of the ZS-OSC problem would enhance the clarity and accessibility of the paper. Following your recommendation, in case the paper is accepted we will update the text in the Methodology section to include a comprehensive definition of ZS-OSC, along with detailed explanations of its inputs, outputs, and significance.
>
> **Explanation regarding the role of the meta-path based network in the context of the task:**
>
> **1. How the Meta-Path-Based Graph Network is Applied in OSC.**
> The meta-path-based graph network plays a central role in modeling the semantic relationships within the knowledge graph (KG) to generate discriminative embeddings for object states. Here is how it operates in the OSC context:
> - KG Construction: The KG is built around the target classes (object states) with nodes representing different concepts, and edges representing semantic relationships such as "is caused by,"  or "is related to."
> - Meta-Path Design: Meta-paths define specific sequences of relationships between nodes in the KG. These paths allow the network to focus on task-relevant relationships.
> - GNN Training: The meta-path-based graph network is implemented using a GNN. The GNN aggregates information along the meta-paths, learning to represent each state node by combining its features with those of its neighbors while prioritizing paths based on their relevance.
> These embeddings are then used as inputs to the classifier, enabling robust recognition of object states even when visual features alone are insufficient.
>
> **2. How the Network is Applied in ZS-OSC.**
> In ZS-OSC, the task extends to recognizing unseen states for which no visual training examples are available. The meta-path-based graph network addresses this challenge by bridging the gap between the semantic knowledge in the KG and the visual features of unseen classes:
> - Semantic Projection into Visual Space: During training, the GNN learns to project semantic embeddings from the KG into the visual space. This is achieved by aligning the KG-generated embeddings for seen states with their corresponding visual representations, effectively "grounding" the semantic information in the visual domain.
> - Embedding 	Generalization: Once trained, the GNN generates embeddings for unseen states by 	leveraging their semantic relationships with  other nodes in the KG.
> - Zero-Shot Classification: The generated embeddings for unseen states are used to classify object states in the visual domain by matching their visual features to the KG-projected embeddings.
>
> **Impact of meta-path:**   Our method has been directly compared against a baseline method that employs a random walk technique for embedding generation. Random walks treat the KG as a homogeneous structure, failing to exploit its inherent heterogeneity. In contrast, our approach leverages meta-paths to explicitly model the diverse relationships encoded in the KG. By keeping all other methodological components identical between the two approaches, we isolate the contribution of meta-paths as the sole critical difference. The significant performance improvements demonstrated by our method across all datasets  can therefore be attributed to the unique ability of meta-paths to exploit heterogeneous information. This finding strongly supports our assertion that meta-path learning effectively introduces and utilizes this heterogeneity. However, we recognize that this evidence could be further strengthened. To address this, we will include qualitative analyses in the revised version of the paper in case of acceptance.
> Regarding comparisons with alternative techniques, as noted in prior responses, we have focused on methods directly applicable to the ZS-OSC task. Random walks were selected as a baseline because they are widely used in KG-based methods. If the reviewer can suggest additional methods that might also be adapted to ZS-OSC, we would be happy to evaluate them in future work.
>
> **Connection between meta-path learning and the zero-shot context:** In our approach, KGs are used to provide the semantic information required to represent target classes that lack visual training examples. However, KGs come with two key challenges:
> - Sensitivity to Noisy or Irrelevant Information: The sheer size and complexity of KGs can introduce noise, making it 	difficult for models to extract meaningful relationships.
> - Underutilization of Heterogeneous Structures: 	Most existing methods treat KGs as homogeneous structures, failing 	to fully exploit their rich, heterogeneous relationships. This leads to embeddings that lack the specificity required for tasks like ZS-OSC.
> In this context, meta-path learning plays a pivotal role in addressing these challenges by *filtering noisy relationships* and *exploiting heterogeneous information*

---

### Official Review · Reviewer_1AuT · 2024-11-02

**Soundness:** 2
**Presentation:** 3
**Contribution:** 1
**Rating:** 5
**Confidence:** 5

**Summary:**

The motivation of this paper is quite unclear. The essence of its method is not much different from the baseline selected by the authors. However, this paper fails to point out what makes it unique compared to other KG-based methods, or to highlight the limitations of other approaches that this paper addresses. Therefore, the motivation of this paper is quite weak, which will significantly impact my final evaluation.

**Strengths:**

The Presentation of this paper is good.

**Weaknesses:**

1. The motivation of this paper is quite unclear. The essence of its method is not much different from the baseline selected by the authors. However, this paper fails to point out what makes it unique compared to other KG-based methods, or to highlight the limitations of other approaches that this paper addresses. Therefore, the motivation of this paper is quite weak, which will significantly impact my final evaluation.

2. While an ablation study is presented, it only explores a narrow set of hyperparameters related to meta-path length, the number of softmax channels, and learning rates. Other critical factors such as embedding dimensions, network depth, and the impact of different types of meta-paths are not examined. Additionally, the study does not provide insights into the underlying reasons for the observed performance changes, limiting its usefulness in understanding the model's behavior.

3. The comparison is limited to a baseline graph-based method and a few large pre-trained models, without including other relevant state-of-the-art methods in zero-shot learning or knowledge graph utilization. The analysis of the results is superficial, lacking in-depth discussion on why the proposed method performs better and under what conditions it might fail.

**Questions:**

Please refer to "Weaknesses".

---

> ### Author Response · Authors · 2024-12-03
> **Response to Reviewer 1AuT**
>
> We thank the reviewer for their valuable feedback and address the raised concerns as follows.
>
> **Motivation of the paper:** The core motivation of this work is to explore the potential of meta-path learning for generating embeddings in the context of zero-shot learning (ZSL). To evaluate our proposed approach, we selected a particularly challenging and meaningful task in Computer Vision: Zero-Shot Object State Classification (ZS-OSC). This task serves as an excellent benchmark to assess the effectiveness of meta-paths in modeling the rich, heterogeneous information embedded in Knowledge Graphs (KGs) and demonstrates their utility in real-world applications where labeled data for unseen classes is unavailable.
> Our method differs fundamentally from the baseline against which it is compared, particularly in how the KG is utilized. While both approaches rely on KGs to learn embeddings, the underlying mechanisms are distinct:
>
>  **1. Meta-Path Learning in Our Approach:**
>  Our 	method leverages meta-paths 	to capture and model heterogeneous relationships within the KG. This 	allows the model to explicitly prioritize task-relevant semantic 	connections (e.g., relationships between object states, actions, and 	attributes) while reducing noise from less relevant relations. By 	assigning weights to different meta-paths, the model dynamically 	adjusts its learning to maximize the usefulness of the relational 	knowledge in the KG.
>
> **2. Random Walks in the Baseline:**
> In contrast, the baseline relies on random walks, which treat the KG as a homogeneous structure. Random walks generate embeddings by sampling paths in an unsupervised manner, which can dilute critical relational patterns and introduce noise, particularly in KGs with 	diverse edge types.
>
>
> To ensure a fair and meaningful comparison, we intentionally kept the other components of our method identical to those of the baseline. This ensures that any observed performance differences can be attributed solely to the utilization of meta-paths. The results, as demonstrated in our experiments, highlight the clear advantage of meta-path learning in capturing the heterogeneous relationships in KGs and significantly improving ZS-OSC performance.
> Finally, to address the concern regarding the uniqueness of our study, it is important to emphasize that this work is the first to:
> - Systematically explore the use of meta-paths for generating embeddings tailored to ZS-OSC.
> - Highlight the limitations of random walks and homogeneous KG modeling in this context.
> - Show the superior effectiveness of meta-path-driven embeddings in producing discriminative representations for unseen object states.
>
> **Ablation study is narrow:**  While we agree that a more extensive ablation study could provide additional insights, our primary objective in this work is to demonstrate the potential of meta-paths for improving embedding generation in the context of zero-shot learning. Since this is, to the best of our knowledge, the first work leveraging meta-paths for Zero-Shot Object State Classification (ZS-OSC), we prioritized presenting the method and comparing its performance to existing approaches over exhaustive hyperparameter exploration.
> That said, your suggestions regarding additional parameters, such as embedding dimensions, network depth, and the impact of different types of meta-paths, are both valuable and well-taken. These factors undoubtedly influence the model’s behavior and performance. However, given the novelty of this work and the focus on introducing meta-paths as a concept, we considered a targeted ablation study (exploring meta-path length, softmax channels, and learning rates) to be sufficient for the scope of this paper. This allows us to isolate and analyze the core contribution of our approach without diluting the primary message.
>
>
>
> **Limited comparison with competing methods:** We have compared our proposed method against all existing methods that support the task of Zero-Shot Object State Classification (ZS-OSC) without requiring adaptation or modification. Specifically, this includes both a graph-based baseline method and several large pre-trained models (LPMs) capable of zero-shot learning, as outlined in Table 4 of the manuscript. These comparisons were selected based on their applicability to ZS-OSC and their ability to handle unseen object states without additional training or task-specific tuning. We recognize, however, that the field of zero-shot learning (ZSL) and knowledge graph (KG) utilization is broad, and it is possible that we may have overlooked methods that could be adapted for this task. If the reviewer could suggest additional state-of-the-art methods relevant to ZS-OSC, we would be happy to evaluate them and include their results in future updates of this work.

---

### Official Review · Reviewer_iVN1 · 2024-11-06

**Soundness:** 2
**Presentation:** 1
**Contribution:** 1
**Rating:** 3
**Confidence:** 4

**Summary:**

This paper aims at zero-shot object state classification (ZS-OSC). The authors propose a novel approach for ZS-OSC that leverages meta-paths to capture complex relationships between object states in a knowledge graph.

**Strengths:**

- Considering the relational knowledge encoded in knowledge graphs for zero-shot object state classification is promising.

**Weaknesses:**

- The writing is poor. The organization of introduction could be improve for a more clear motivation. The task of ZS-OSC could be first introduced.
- It could be better to first present the comparison results in the section of experiments.
- No other methods for exploiting the relational information within KGs (such as filtering mechanisms, ad-hoc KG construction, or random walks) are compared
- The improvement is not obvious. Besides, did the authors run the experiments multiple times to obtain the mean results?

**Questions:**

Please see the weaknesses.

---

> ### Author Response · Authors · 2024-12-03
> **Response to Reviewer iVN1**
>
> We thank the reviewer for their valuable feedback and address the raised concerns as follows.
>
>
> **Lack of comparison with other KG-based methods:** The main objective of this work is to assess the potential of using meta-paths in the challenging task of Object-Agnotic Zero-Shot Object State Classification (ZS-OSC). To the best of our knowledge, the only existing approach specifically addressing ZS-OSC is the baseline method against which we have compared our work~\citep{gouidis2023leveraging}. This baseline relies on random walks to extract relational information from the knowledge graph (KG). We emphasize that ZS-OSC is a highly underexplored problem, distinct from other zero-shot tasks such as Compositional Zero-Shot Learning (CZSL), which require explicit object class information, a constraint that does not apply to ZS-OSC. As such, existing methods in related domains (e.g., CZSL or attribute-based zero-shot learning) are not directly applicable here. If the reviewer is aware of specific methods that align with the requirements of Object-Agnotic ZS-OSC and operate under its constraints, we would greatly appreciate pointers to such approaches.
>
>
>
>
>
>
>
> **Not obvious improvement:**  We believe that the improvements achieved through the use of meta-path learning are evident from the results presented in Table 4 on page 9 of the manuscript. Specifically, our method demonstrates significant performance gains compared to all competing methods across all datasets. For instance, our approach outperforms both the baseline method (Gouidis et al., 2023) and state-of-the-art (SoTA) large pre-trained models (LPMs), with improvements highlighted in the table’s bottom row. These gains reflect the effectiveness of leveraging meta-paths to capture complex semantic relationships within knowledge graphs, which directly enhance the quality of the generated embeddings. Furthermore, our meta-path-based approach significantly improves the baseline method, which relies on random walks. This demonstrates the value of explicitly modeling heterogeneous relationships in knowledge graphs rather than treating all connections uniformly.
>
> **Reported experimental values:** Regarding the reported experimental values, they represent the mean results obtained from 10 independent runs, with each run corresponding to a specific set of hyperparameters.

---

### Note · Authors · 2025-02-25

I have read and agree with the venue's withdrawal policy on behalf of myself and my co-authors.

---

### Meta-Review · Area_Chair_gSBR · 2024-12-21

**Metareview:**

This paper proposes to learn to project semantic information from the KG into the visual space via meta-path learning, generating discriminative visual embeddings for unseen state classes, for Zero-Shot Object State Classification (ZS-OSC). The authors have provided a rebuttal, and all reviewers have checked it and recommended rejection. After careful consideration, the Area Chair (AC) agreed with the reviewers' assessments and decided to reject the paper.

**Additional Comments On Reviewer Discussion:**

After carefully checking the rebuttal and feedback, a rejection decision has been made by AC.

---

### Decision · Program_Chairs · 2025-01-22

Reject